# Sea Surface Salinity Anomaly in the Bay of Bengal during the 2010 Extremely Negative IOD Event

Shuling Chen [1,2], Jing Cha [1,2], Fuwen Qiu [1,2,*], Chunsheng Jing [1,2], Yun Qiu [1,2,3,4] and Jindian Xu [1,2]

1   The Third Institute of Oceanography, Ministry of Natural Resources, Xiamen 361005, China
2   Fujian Provincial Key Laboratory of Physical and Geological Processes, Xiamen 361005, China
3   Southern Marine Science and Engineering Guangdong Laboratory (Zhuhai), Zhuhai 519000, China
4   Laboratory for Regional Oceanography and Numerical Modeling, Qingdao National Laboratory for Marine Science and Technology, Qingdao 266000, China
*   Correspondence: qiufuwen@tio.org.cn

**Abstract:** Based on Soil Moisture and Ocean Salinity (SMOS) data and the Ocean Reanalysis System 5 (ORAS5) dataset, positive salinity anomalies exceeding 2 psu in the northern Bay of Bengal (BoB) and negative salinity anomalies with the peak of the freshening anomalies reaching −2 psu around Sri Lanka were observed in autumn 2010. Here, an analysis of the anomalous salt budget revealed that anomalous horizontal advection contributed most to the variability in salinity in the BoB. With the development of La Niña and negative Indian Ocean dipole (nIOD) in summer and autumn, the strong summer monsoon current and Wyrtki jet combined with the anomalous basin-scale cyclonic circulation led to more high-salinity water entering the northern BoB. In addition, more freshwater was transported southward along the eastern coast of India by east Indian coastal current (EICC) in autumn, resulting in extremely negative salinity anomalies around Sri Lanka and positive salinity anomalies in the northern BoB. Moreover, the freshwater around Sri Lanka was carried farther into the southeastern Arabian Sea by the west Indian coastal current (WICC) in November, which affected the salinity stratification in winter and then influenced the variation of the Arabian Sea Mini Warm Pool (ASMWP) in the following spring. The ASMWP could affect the Indian summer monsoon (ISM) through its influence on the monsoon onset vortex (MOV) over the southeast Arabian Sea (SEAS).

**Keywords:** sea surface salinity 1; the Bay of Bengal 2; anomalous salt budget 3

## 1. Introduction

The Bay of Bengal (BoB) is a semi-enclosed basin in the northeastern Indian Ocean and features low sea surface salinity (SSS) all year round [1], which is mainly due to the excessive precipitation (P) minus evaporation (E). Particularly, the frequent occurrence of extreme rainfall over the BoB and the runoff of rivers result in a large decrease in SSS during the summer monsoon (June to September) [2].

Salinity variability during and after the summer monsoon could influence the stratification of the upper ocean, further modulate the formation of barrier layers in winter, and impact the air–sea interactions in the following spring, so it is important to monitor the salinity variation in the BoB [3,4]. Many studies have been carried out to determine the primary factors acting on salinity variability in the BoB during and after the summer monsoon season. For instance, Wang et al. [5] used a box model based on the Regional Ocean Modeling System (ROMS) and found that the river runoff and precipitation are the dominant factors modulating the salinity variability in the BoB during the summer monsoon, which is consistent with previous studies [2,6]. Due to the excessive rainfall and massive river runoff, the SSS in the northern BoB sharply decreases off shore [2]. Meanwhile, the summer monsoon current (SMC) carries high-salinity water from the Arabian Sea (AS) to balance the salinity of the BoB [7,8]. After the summer monsoon season (October to November), the salinity variation is mostly due to horizontal advection [5]. In October,

the east India coastal current (EICC) transports freshwater southward along the eastern coast of India from the northern bay, reaching the southern tip of India in November [2]. At the same time, driven by westerly winds, the Wyrtki jet carries high-salinity waters and flows eastward along the equator to the western coast of Sumatra, where part of the currents turn northward into the BoB [9,10].

On the interannual timescale, the Indian Ocean dipole (IOD) and the El Niño–Southern Oscillation (ENSO) have been identified as the dominant causes of salinity variability in the BoB [11,12]. SSS variability in the BoB is mainly controlled by local freshwater flux (P-E) in summer and horizontal advection in autumn, which is associated with IOD and ENSO [13,14]. The variability of the Indian summer monsoon (ISM) is related to IOD and ENSO [13]. The positive IOD (pIOD) is associated with strong ISM with above normal rainfall in the central part of India and below normal rainfall to the north and south of it. In contrast, the negative IOD is associated with reduced ISM with above (below) normal rainfall in western (eastern) India [15]. El Niño is accompanied by reduced ISM due to anomalous subsidence while La Niña enhances moisture convergence over India and causes rainfall surplus [16,17]. Accordingly, when IOD events co-occur with ENSO, the impact of pIOD (nIOD) on rainfall over India could counteract with that of El Niño (La Niña) [18].

In addition, remote forcing from IOD and ENSO also has an impact on the surface circulation which dominates the distribution of SSS in the BoB after the summer monsoon [19–21]. The EICC and Wyrtki jet are strongly influenced by the IOD [9,22]. During the peak phase of the nIOD (pIOD) from September to November, downwelling (upwelling) Kelvin waves propagate eastward along the equator, which are attributed to anomalous westerlies (easterlies). After reaching the western coast of Sumatra, equatorial downwelling (upwelling) Kelvin waves turn into coastal downwelling (upwelling) Kelvin waves accompanied by a positive (negative) sea level anomaly (SLA), while the energy is trapped by the coast [23,24]. The equatorial downwelling (upwelling) Kelvin waves strengthen (weaken) the Wyrtki jet, while the coastal downwelling (upwelling) Kelvin waves intensify (weaken) the EICC, which changes the distribution of salinity in the BoB in autumn [22,25]. As nIOD (pIOD) events co-occur with La Niña (El Niño) events, the anomalous westerlies (easterlies) strengthen, which intensifies the associated possesses, resulting in strong (weak) EICC and Wyrtki jet.

The year 2010 was notable for its extreme warming, with the average global mean surface temperature among the two warmest years during 1880–2010. In addition, several climate modes interacted with each other in 2010, causing complex climate and weather variations globally. For India, the annual mean temperature was 0.93 °C higher than the average temperature during 1961–1990. Severe heat wave events over India in May claimed hundreds of lives. A severe cyclonic storm Laila formed over the southeast BoB, causing widespread damage and resulting in more than 50 deaths [26]. The climate changes and frequent occurrence of extreme weather events make the year 2010 remarkable.

In addition, an extreme nIOD event co-occurred with a moderate-to-strong La Niña event during late boreal summer to fall in 2010 (Figure 1), leading to massive westerly anomalies in the equatorial Indian Ocean [26]. Eastward downwelling equatorial Kelvin waves driven by westerly anomalies reached the western coast of Sumatra and converted into coastal downwelling Kelvin waves, which associated with currents along the coastal BoB. The SSS in the BoB is modulated by currents related to remote forcing from the equatorial Indian Ocean [21], so it is necessary to analyze the variation in the SSS distribution in summer and autumn 2010. Interestingly, a dramatic salinity anomaly distribution in the BoB was observed in autumn 2010. However, the relative role of the processes (e.g., the freshwater flux and anomalous advection) in contributing to the SSS distribution in 2010 have not yet been clearly discussed. To achieve the goal mentioned above, this paper analyzes the evolution of the salinity anomalies in the BoB and explores the associated processes. The organization of the paper is as follows. Data sources and methodologies are presented in Section 2. In Section 3, a description of the characteristics of the salinity

anomaly and an analysis of the associated processes are given. Section 4 presents the impact of remote forcing from the equatorial Indian Ocean on salinity variation as well as the impact of the salinity anomaly in autumn 2010 on the summer monsoon in 2011. Section 5 presents the conclusion.

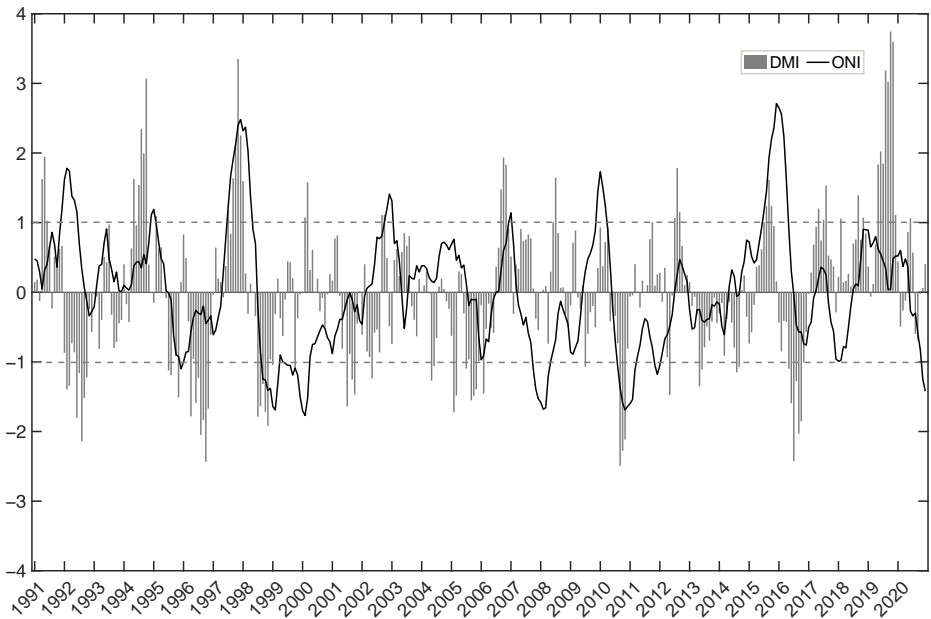

**Figure 1.** Dipole Mode Index (DMI) time series (gray bar) and Niño 3.4 index (ONI) time series (black line).

## 2. Data and Methods

### 2.1. Data

The Soil Moisture and Ocean Salinity (SMOS) version 3 Level 3 SSS product was obtained from L'OCEAN Centre Aval de Traitement des Données SMOS [27]. SMOS is available at 0.25° gridded horizontal resolution from 2010 to the present, with data for every 4 days. Akhil et al. [20] showed that the SMOS SSS compares well with in situ observations and satellite products (including Soil Moisture Active and Passive (SMAP) and Aquarius) over the BoB in 2010–2019.

The surface currents were provided by the Ocean Surface Current Analyses Real-time (OSCAR) data set and taken on global 1/3 degree grid with a 5-day resolution [28].

The monthly Ocean Reanalysis System 5 (ORAS5) dataset was obtained from the European Centre for Medium-Range Weather Forecasts (ECMWF) [29]. ORAS5 is based on the Nucleus for European Modelling of the Ocean (NEMO), forced by both observations (HadISST2 SST, Operational Sea Surface Temperature and Ice Analysis (OSTIA) sea-ice concentration (SIC), in situ temperature and salinity profiles, Archiving Validation and interpretation of Satellite Oceanography (AVISO) DT2014 Sea level anomaly (SLA)) and forcing fields. No observations from SMOS or other satellite measures of SSS are used in the model, so its output is independent from the SMOS observations. ORAS5 has a spatial resolution of $0.25° \times 0.25°$ with 75 vertical depth levels and is available from 1958 to the present with a monthly resolution. The SSS, mixed layer depth (MLD), zonal and meridional velocity and salinity at different levels were obtained from ORAS5 and used for calculating the anomalous salt budget. The MLD, the depth of the ocean where the average sea water density exceeds the near surface density plus 0.03 kg/m³, was used to calculate the mixed layer averaged salinity.

The monthly mean ERA5 reanalysis product is available from ECMWF from 1959 to the present [30] with a spatial resolution of $0.25° \times 0.25°$. The variables we used in this study were evaporation, total precipitation, runoff, zonal (u) and meridional (v) components of wind at 850 hPa, and moist static energy. Mahto, S. S. and Mishra, V. [31] evaluated new-ERA reanalysis products (including ERA-5, Climate Forecast System Reanalysis, ERA-

Interim, Modern Era Retrospective Analysis for Research and Applications version 2, and Japanese 55-year Reanalysis Project) and found that the ERA-5 performs better than the other reanalysis products and can be used for hydrologic assessments in India.

*2.2. Methods*

2.2.1. Anomalous Mixed Layer Salt Budget

The anomalous mixed layer salt budget was calculated to explore the dominant processes contributing to salinity anomalies in the mixed layer. Previous studies calculated the mixed layer salt budget and suggested that the local freshwater flux and horizontal advection are the dominant mechanisms of SSS variability in the tropical Indian Ocean [11,32]. The anomalous mixed layer salt budget was calculated using the following formula according to previous studies [33]:

$$\frac{\partial S'}{\partial t} = \overline{S_0}\frac{(E-P)'}{h} - \overline{u}\frac{\partial S'}{\partial x} - \overline{v}\frac{\partial S'}{\partial y} - u'\frac{\overline{\partial S}}{\partial x} - v'\frac{\overline{\partial S}}{\partial y} - u'\frac{\partial S'}{\partial x} - v'\frac{\partial S'}{\partial y} + R \tag{1}$$

where overbar terms represent the climatological mean seasonal cycle and primed terms represent anomalies. S and $S_0$ are the mixed layer salinity and sea surface salinity, respectively. The variable h is the MLD, and u (v) is the mixed layer zonal (meridional) current velocity. E is evaporation, and P is precipitation. The values of h, S, $S_0$, and u (v) are from ORAS5, while E and P are from ERA5. The residual R represents physical processes such as runoff and vertical mixing that cannot be estimated directly. The terms in Equation (1) from left to right are the anomalous salinity tendency, anomalous sea surface freshwater flux of the climatological salinity, zonal advection of anomalous salinity by the climatological current, meridional advection of anomalous salinity by the climatological current, zonal advection of climatological salinity by the anomalous current, meridional advection of climatological salinity by the anomalous current, zonal advection of anomalous salinity by the anomalous current, meridional advection of anomalous salinity by the anomalous current, and residual term. The horizontal advection of anomalous salinity by the climatological current is ADV-SA = $-\overline{u}\frac{\partial S'}{\partial x} - \overline{v}\frac{\partial S'}{\partial y}$; the horizontal advection of climatological salinity by the anomalous current is ADV-UVA = $-u'\frac{\overline{\partial S}}{\partial x} - v'\frac{\overline{\partial S}}{\partial y}$; and the horizontal advection of anomalous salinity by the anomalous current is ADV-UVASA = $-u'\frac{\partial S'}{\partial x} - v'\frac{\partial S'}{\partial y}$. The horizontal advection term contains ADV-SA, ADV-UVA, and ADV-UVASA. We used the anomalous mixed layer salt budget to analyze the contribution of horizontal advection and freshwater flux to SSS variability.

2.2.2. Anomalous Transports of Salinity and Freshwater

Salinity and freshwater transports in the mixed layer were calculated using the following equations:

Meridional salinity transport:

$$F_{S,v} = \int_0^L \rho v(x)s(x)h(x)dx \tag{2}$$

Meridional mass transport:

$$F_{M,v} = \int_0^L \rho v(x)h(x)dx \tag{3}$$

Meridional freshwater transport:

$$F_{FWv} = F_{M,v} - F_{S,v} \tag{4}$$

where $\rho$ is the seawater density; v is the mixed-layer-averaged meridional current velocity; dx is the difference between longitudinal grid points; s is the mixed layer averaged

salinity; h is the mixed layer depth (MLD); and L is the zonal distance of the section considered [32,34–36]. The anomalous transport was then calculated by subtracting climatological values from the real transports above.

### 2.2.3. Moist Static Energy (MSE)

The convective activity over a region can be measured by the MSE [37,38], which is defined as

$$MSE = gz + C_pT + Lq \tag{5}$$

where $g = 9.80665$ m s$^{-2}$ is the gravitational constant, z is geopotential height from ERA5, $C_p = 1004.6$ JK$^{-1}$kg$^{-1}$ is the heat capacity at constant pressure, T is temperature in the atmosphere from ERA5, $L = 2.507 \times 10^6$ J kg$^{-1}$ is the latent heat of vaporization of water from ERA5, and q is specific humidity.

## 3. Results

### 3.1. Distribution of Salinity Anomalies for Summer and Autumn 2010

The distribution of climatological SSS in the northern Indian Ocean in summer (from June to September) and autumn (October and November) is demonstrated first (Figure 2). This figure shows that the average salinity in the AS is approximately 3 psu higher than that in the BoB from summer to autumn. Unlike the relatively uniform distribution of SSS in the AS, low-salinity waters in the BoB are confined to coastal areas, while a high-salinity tongue extends to the east of Sri Lanka. Moreover, the low-salinity waters extend anticlockwise from the northern and eastern coastal regions over time. In November, the extension of low-salinity waters is as far as south of Sri Lanka. On the other hand, the high-salinity waters can be carried to the center of the BoB due to the strong current near the equator, corresponding to the intrusion of high-salinity waters from the AS. These observations suggest that the distribution of SSS in the BoB is highly correlated with the currents, including the SMC, EICC and the Wyrtki jet [4,10,39].

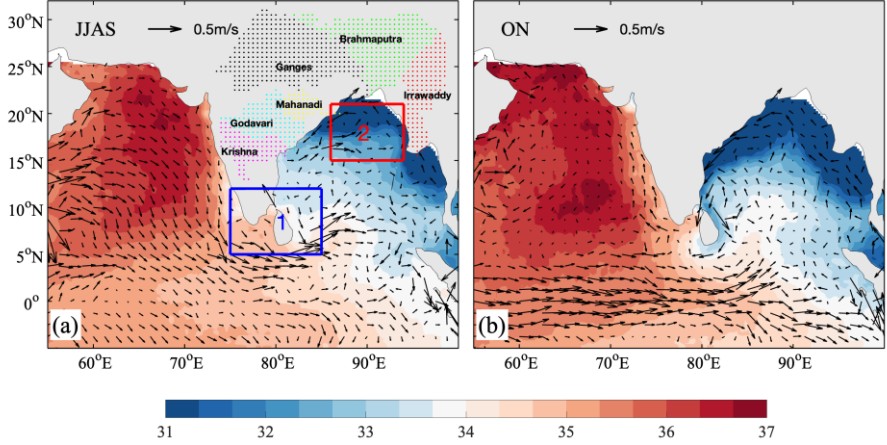

**Figure 2.** Composites of the distribution of SMOS sea surface salinity (shaded; psu) and OSCAR currents (vectors; m/s) (**a**) from June to September, (**b**) in October and November for 2010–2019. The blue box indicates region 1 around Sri Lanka (75–85°E, 5–12°N), and the red box indicates region 2 in the northern BoB (86–94°E, 15–21°N). The distribution of rivers is marked with dots (red, green, white, yellow, blue, and pink represent Irrawaddy, Brahmaputra, Ganges, Mahanadi, Godavari, and Krishna, respectively).

Two regions (region 1: 75–85°E, 5–12°N; region 2: 86–94°E, 15–21°N) were chosen to check the reliability of the results depending on different datasets (Figure 2). Region 1 (region 2) corresponds to significant negative (positive) salinity anomalies (Figure 3). The time series of the average SSS of the two regions based on the SMOS and ORAS5 datasets are illustrated in Figure 4. The salinity from the two datasets shows similar seasonal variations

with the lowest salinity in autumn. On average, the salinity in the northern BoB is much lower than that around Sri Lanka due to river runoff. In general, the basic characteristics are captured by both datasets. Therefore, the salinity of the two regions based on the two datasets are found to be consistent with each other, suggesting that the results of salinity variability are reliable. In addition, Roman-Stork et al. [35] also compared SSS from Satellite-derived (including SMOS, SMAP, and Aquarius) and the reanalysis products (including SODA and ORAS5) for the eastern Arabian sea and BoB. By comparison, ORAS5 shares a spatial pattern very close to the satellite products.

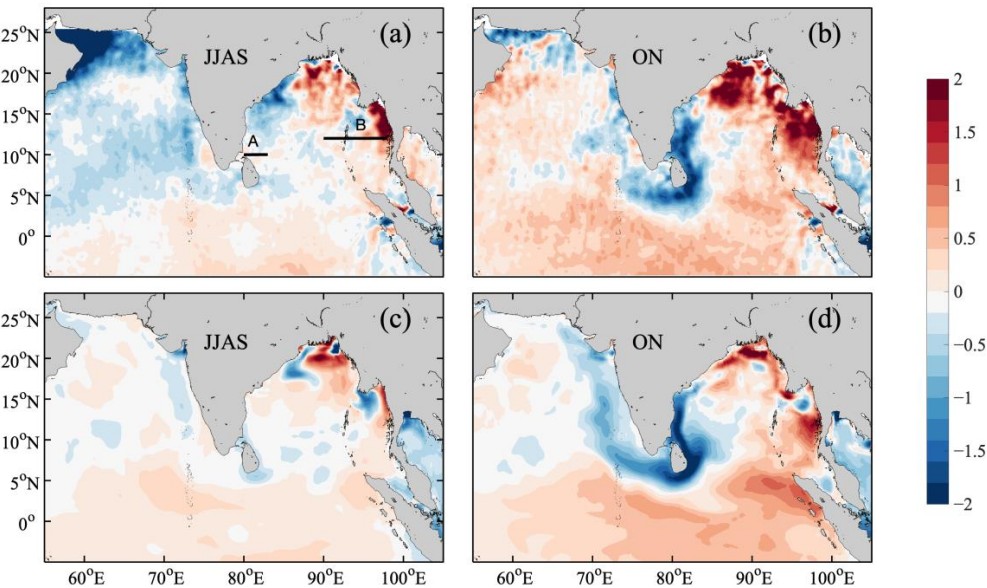

**Figure 3.** Sea surface salinity anomalies of SMOS (**a**,**b**) and ORAS5 (**c**,**d**) in 2010 relative to the mean for the period 2010–2019. The lines in (**a**) indicate section A (80–83°E, 10°N) and section B (90–98°E, 12°N).

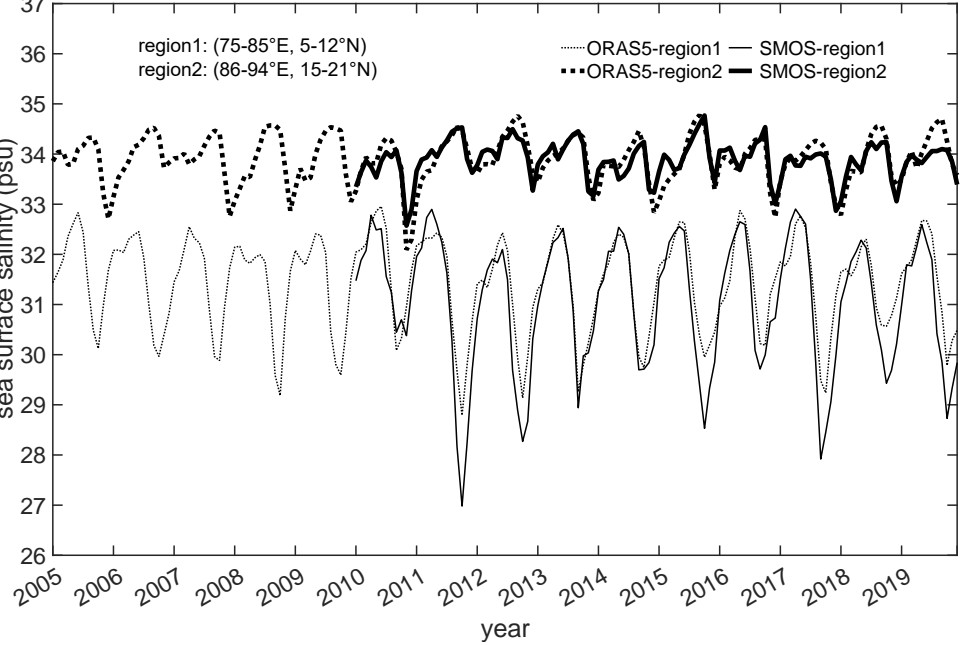

**Figure 4.** Comparison of the time series of ORAS5 and SMOS sea surface salinity for region 1 (thin line) and region 2 (thick line) from 2005 to 2019. Dash lines represent the sea surface salinity from the ORAS5 dataset; solid lines represent the sea surface salinity from the SMOS dataset.

The SSS anomalies in summer and autumn 2010 from the two datasets are illustrated in Figure 3. The anomaly is obtained by subtracting the climatological cycle values from the values in 2010. It is clear that both datasets describe a dipole structure of salinity anomalies in the northern BoB in summer. In autumn, significant positive SSS anomalies exceeding 2 psu exist in the northern BoB (86–94°E, 15–21°N) and negative SSS anomalies with the peak of the freshening anomalies reaching -2 psu exist east and south of Sri Lanka (75–85°E, 5–12°N). The low-salinity anomalies originate from the northern coast of the BoB in summer and move southward along the eastern coastline of India. There are differences between the SSS values from the two datasets, especially the negative anomalies in the northern AS. In summer 2010, the SSS anomalies in the AS from ORAS5 were less significant, while there was remarkably low salinity in the northern coastal areas based on SSS from SMOS. Although differences appeared in the magnitude of salinity anomalies, the evolution of salinity anomalies in the BoB described by the two datasets are consistent.

### 3.2. Mixed Layer Anomalous Salt Budget

An analysis of the anomalous salinity budget in the mixed layer was carried out to determine the dominant processes. The spatial patterns of anomalous salinity tendency (TEND), anomalous local sea surface freshwater flux (FWF) and anomalous horizontal advection (ADV) are shown in Figure 5. Overall, the horizontal advection was similar in magnitude (over ±1.5 psu) to that of the salinity tendency in the northern BoB. The distribution of the positive salinity anomalies on the northern coast of the BoB and the freshwater transported to the eastern coast of India and Sri Lanka was well delineated by horizontal advection in autumn 2010, which was consistent with Figure 6. As shown in Figure 6, the positive salinity anomaly appeared in the northern BoB in July and transported southward slightly in autumn 2010, while the negative salinity anomaly in the northwest coast of India in summer moved southward in October. By contrast, the freshwater flux contributed little to the salinity tendency, indicating that the anomalous salinity is determined by horizontal advection.

The two regions chosen in Figure 2a showed large variations in salinity in the BoB. Therefore, the anomalous salinity budget averaged in the two regions was used to examine the related processes in 2010 (Figure 7). In the coastal areas around Sri Lanka, the significant negative salinity tendency in autumn was mainly attributed to horizontal advection, especially the horizontal advection of the climatological salinity by the anomalous current in October. In November, both the climatological salinity by the anomalous current and the anomalous salinity by the anomaly current contributed to the negative salinity anomaly around Sri Lanka. Although the magnitude of anomalous local sea surface freshwater flux term was much smaller, this process also played an important role in the salinity variation in summer. In addition, the freshwater flux led to the negative salinity tendency around Sri Lanka in August, September, and November but to a lesser extent. The magnitude of salinity tendency around Sri Lanka was relatively small except in November, probably due to the limitation of the monthly temporal resolution of the data. For the salinity in the northern BoB, there was a remarkable positive tendency resulting from the horizontal advection of the salinity climatological salinity by the anomalous current in October. The freshwater flux, however, had the reverse effect on the positive salinity tendency in October. In November, the magnitude of the salinity tendency was small, but the horizontal advection of the climatological salinity by the anomalous current term contributed to the positive salinity anomaly in the northern BoB. These observations suggest that the variability of surface currents was vital for the salinity distribution in both the region around Sri Lanka and the northern BoB in autumn 2010. In the following section, we perform a further analysis of the anomalous currents based on the ORAS5 dataset.

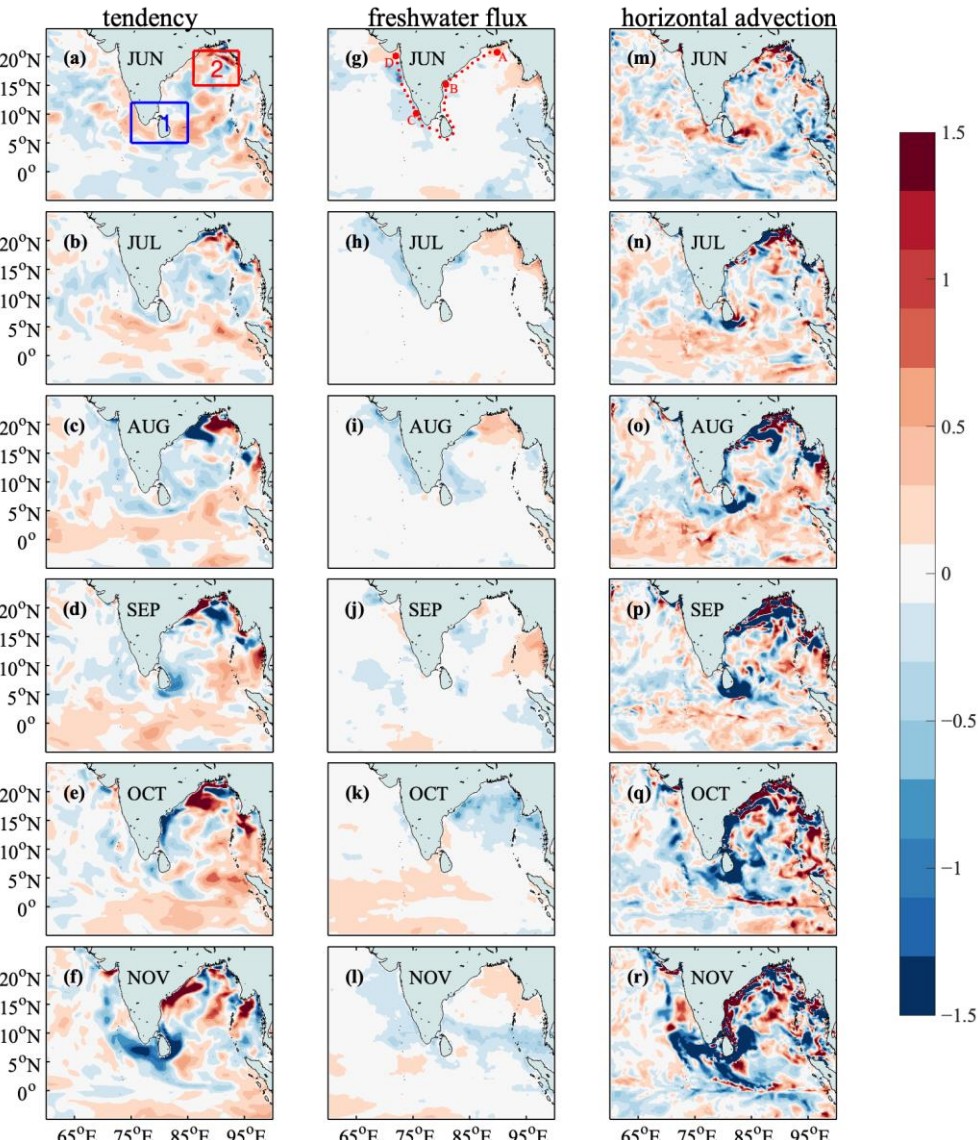

**Figure 5.** Monthly mixed layer anomalous salt budget in 2010 (shaded; psu/month), including the tendency term (**a–f**), anomalous local freshwater flux term (**g–l**) and horizontal advection term (**m–r**). The blue box indicates region 1 around Sri Lanka (75–85°E, 5–12°N), and the red box indicates region 2 in the northern BoB (86–94°E, 15–21°N). The red dots in (**g**) indicate the track of low-salinity water.

### 3.3. Anomalies of Currents in 2010

The mixed layer salinity and current climatology (a–c) and anomaly (d–f) in 2010 summer and autumn with respect to the period 2010–2019 are given in Figure 8. There had been climatological eastward currents along the equator since summer. The eastward currents brought waters from the AS to the southern BoB, which was the so-called intrusion of high-salinity waters. The SMC in summer and the Wyrtki jet in October of 2010 flowed farther eastward, suggesting a stronger intrusion of high-salinity waters. Accompanied by this intrusion, the anomalous cyclonic circulation in the BoB favored water transport to the northern coast of the Indian Ocean and led to the positive salinity tendency there. On the other hand, the strengthened EICC brought the local low-salinity waters toward the eastern coast of Sri Lanka. The transport of low-salinity waters is a climatological process but was strengthened significantly and finally led to the anomalous low salinity around Sri Lanka and the extremely high salinity in the northern BoB in autumn 2010.

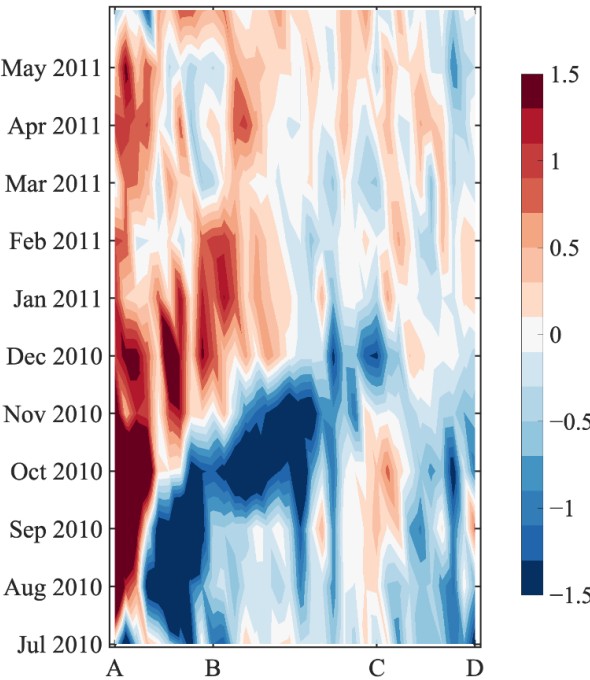

**Figure 6.** Hovmoller diagram of SMOS sea surface salinity anomaly of the dots along the east and west coasts of India in Figure 5g from July 2010 to May 2010 relative to the mean for the period 2010–2019.

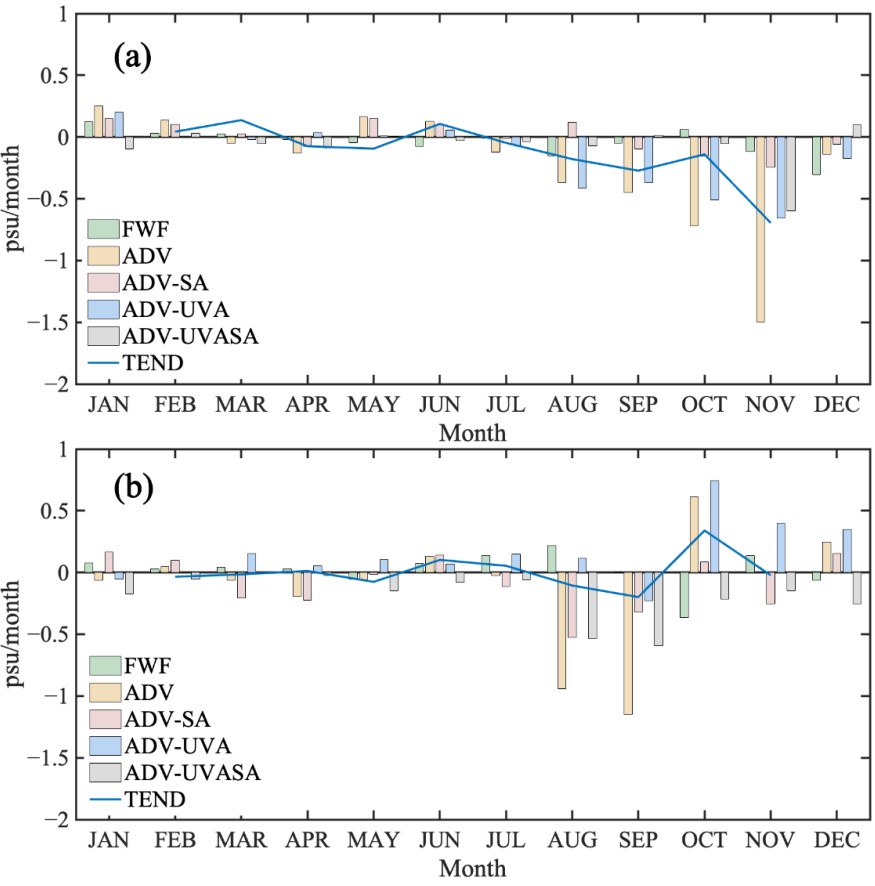

**Figure 7.** Anomalous salt budget terms for (**a**) region 1: the Sri Lanka region (75–85°E, 5–12°N) and (**b**) region 2: the northern BoB region (86–94°E, 15–21°N) in 2010.

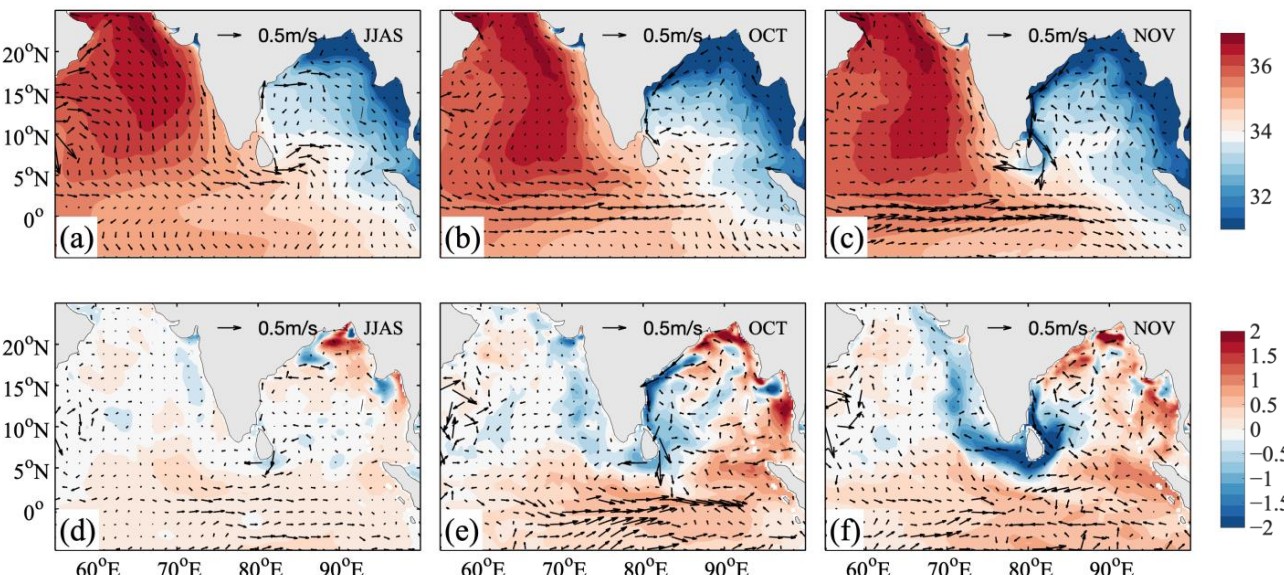

**Figure 8.** Mixed layer salinity and current climatology (**a**–**c**) and anomaly (**d**–**f**) in 2010 summer and autumn from ORAS5.

To quantify the freshwater transported southward along the eastern coast of India to Sri Lanka and the high-salinity water transported northward into the northern BoB, two sections (section A: 80–83°E, 10°N; section B: 90–98°, 12°N) were selected to calculate and compare the transports in 2010. In general, the southward freshwater transport in section A and the northward salinity transport in section B (Figure 4a) within the mixed layer are presented in Figure 9. A positive (negative) value denotes anomalous northward (southward) transport. Note that the value of total freshwater transport east of Sri Lanka was more than $3 \times 10^{10}$ kg/s from August to October 2010 (Figure 9c), while the climatological transport was much smaller, especially in September and October. The anomalous transport east of Sri Lanka in 2010 amplified the climatological freshwater transports (Figure 9), resulting in the large magnitude of negative salinity anomalies around Sri Lanka in 2010 autumn. In contrast, the anomalous salinity transport in the northern BoB was comparable to the magnitude of the climatological salinity transport (approximately $4 \times 10^8$ kg/s every month) but in the opposite direction. This indicates that the anomalous salinity transport in the eastern BoB in 2010 had a countereffect on the climatological salinity transport and resulted in more high-salinity waters being carried into the northern BoB from July to October.

### 3.4. Precipitation and River Discharge Anomalies in 2010

Although the anomalous local freshwater flux contributed less to the salinity variations in the two regions, we still analyzed the related precipitation and runoff around the BoB in 2010 to check the influence of the summer monsoon. In summer 2010, the monsoon winds were rather weak compared to the climatological winds (Figure 10). The anomalous easterlies prevailed over the northern BoB, resulting in less precipitation and less runoff along the northern coast of the BoB, which finally led to the positive salinity tendency in the northern BoB (Figure 5). In October, however, the westerly anomalies and anomalous cyclonic circulation contributed to the increased precipitation in the northern BoB, which played a negative role in the total salinity tendency there (Figure 5k). In eastern India, on the other hand, the anomalous easterlies favored more precipitation in summer, which caused remarkable runoff on the eastern coast of India. The average volume of runoff was ~$10^4$ m³/s during the summer monsoon [40,41], including the runoff from the Mahanadi, Godavari and Krishna rivers. In the summer of 2010, the anomalous volume of runoff was as much as the climatological mean value, especially in September, up to $1.5 \times 10^4$ m³/s (Figure 11). The freshwater was transported to the east of Sri Lanka later in autumn,

resulting in a negative salinity tendency. The anomalous volume of runoff in the northern BoB (including the Ganges, Brahmaputra and Irrawaddy rivers) in June, September, and October was as much as that on the eastern coast of India (Figure 11) but much less than the climatological value (~$12 \times 10^4$ m$^3$/s) [42,43]. Therefore, the volume of runoff in the northern BoB plays a minor role in salinity variation.

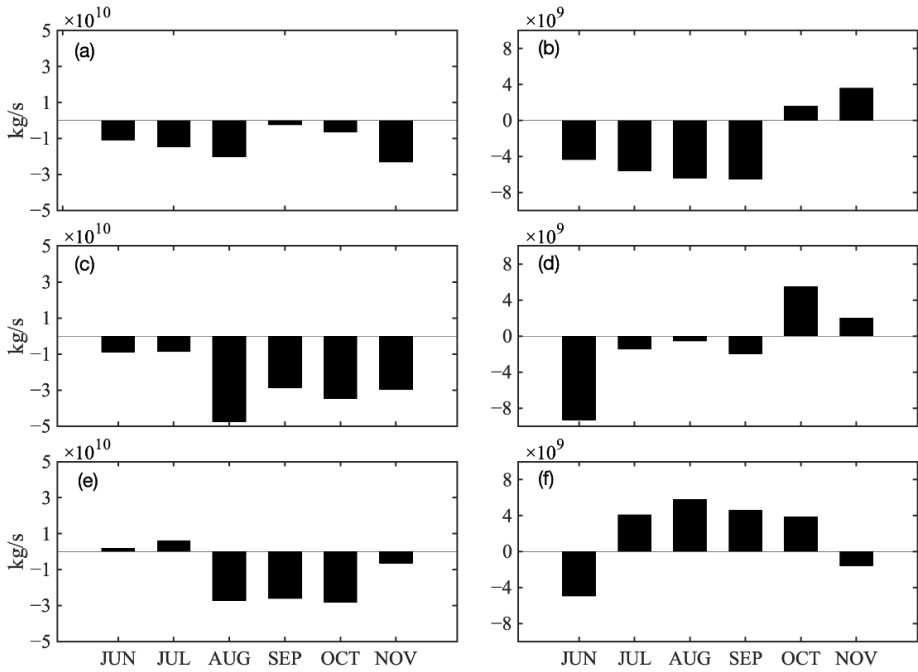

**Figure 9.** The freshwater transport in section A (80–83°E, 10°N; **a,c,e**) and salinity transport in section B (90–98°E, 12°N; **b,d,f**) marked in Figure 3 from June to November. The panels from top to bottom correspond to the climatological transports, transports in 2010 and anomalous transports in 2010 (the second panel minus the first panel).

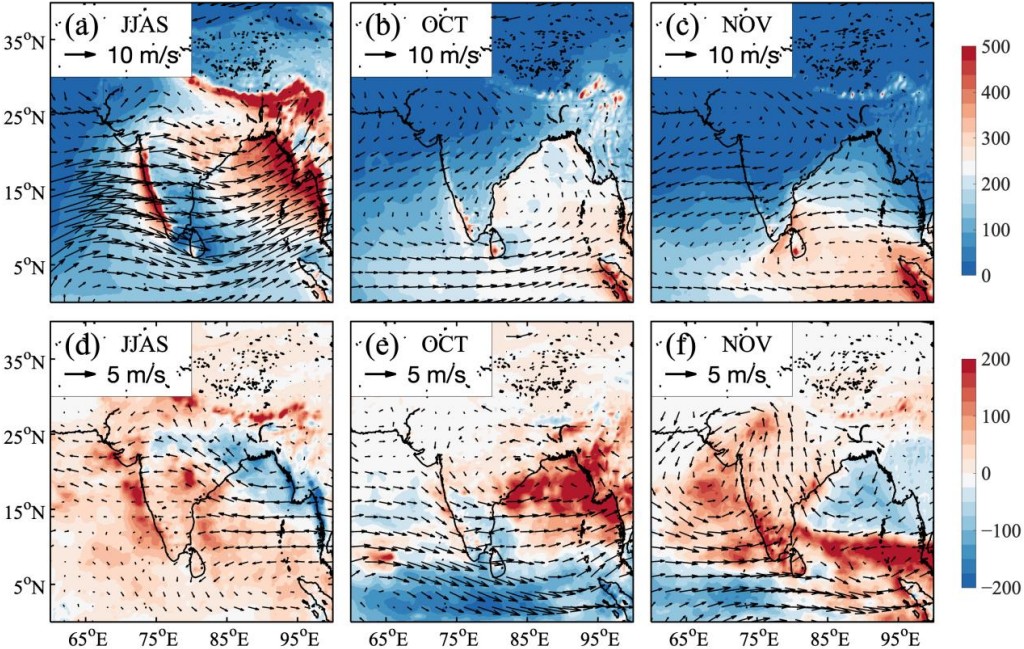

**Figure 10.** Horizontal wind at 850 hPa (vector; m/s) and precipitation (shaded; mm) climatology (**a–c**) and anomaly (**d–f**) in 2010 from summer monsoon season to November based on ERA5 data.

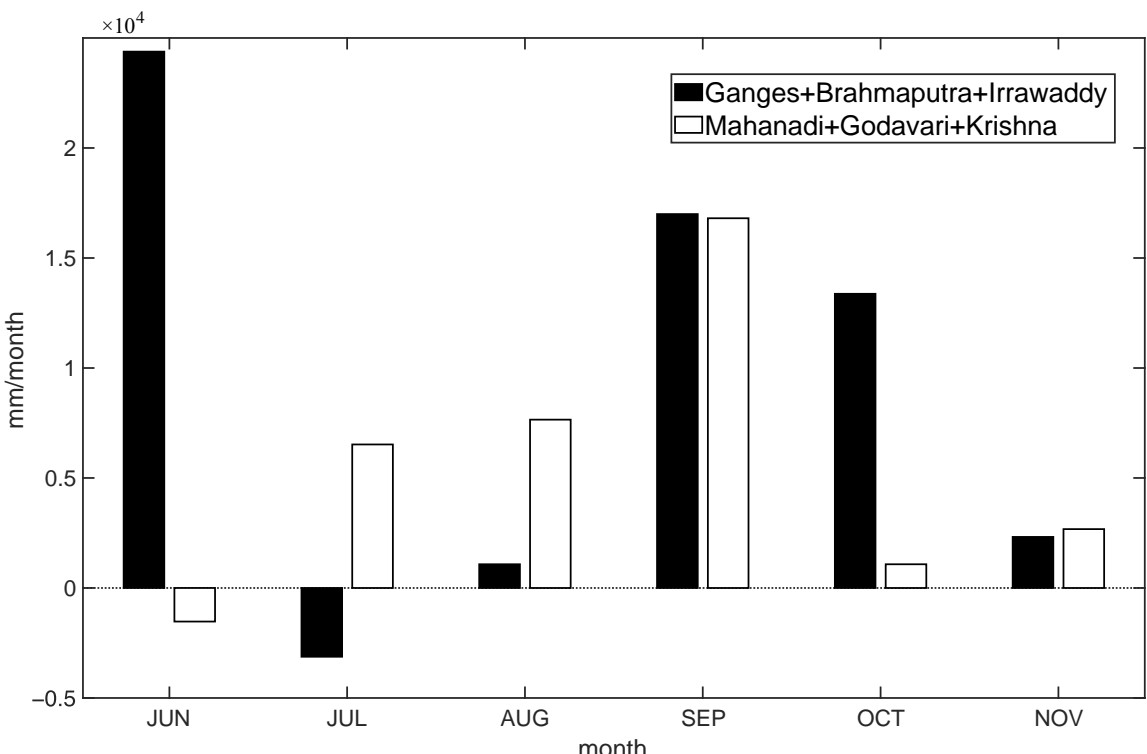

**Figure 11.** The anomalous runoff along the west (white bars) and north (dark bars) coast of the BoB from June to November 2010.

## 4. Discussion

The salinity stratification of the upper ocean is thought to play an important role in local mixed layer thermodynamics and air–sea interactions, so it is important to monitor the salinity variation in the BoB. Many studies have been carried out to explore the causes of salinity variation in different parts of the BoB. Pant et al. studied near-surface salinity in the middle part of the BoB during 2005–2013 and pointed out that the oceanic and atmospheric conditions associated with IOD rather than ENSO control the interannual variability of salinity in the BoB [14]. Li et al. [21] observed the extremely low salinity anomaly in the BoB in 2012 and explored the main cause of this event. They suggested that oceanic circulation related to the rare combination of pIOD and La Niña dominated the event. Chatianya et al. [6] gathered six different in situ datasets to explore the SSS interannual variability in the BoB for the period of 2009–2014. They found abnormal salinification over the northeastern BoB from September–November 2010, which is consistent with our study. However, the monthly processes associated with such variation have not been discussed before. With the co-occurrence of nIOD and La Niña, more analyses of the monthly processes of the formation of this unusual distribution of salinity anomalies in the BoB in autumn 2010 are needed. In this study, we used the SMOS dataset and reanalysis products to detect the unusual distribution of positive salinity anomalies in the northern BoB and negative salinity anomalies east of Sri Lanka in autumn 2010 (Figure 3b,d). We analyzed the anomalous salinity budget and determined that anomalous horizontal advection played a dominant role in the salinity fluctuation within the mixed layer in autumn 2010.

Current anomalies were predominant in producing the salinity fluctuations in the BoB. Webber et al. [44] and Li et al. [45] showed that the strength and position of the SMC are driven by local forcing (wind stress curl over the Sri Lanka Dome) and remote forcing from the equator (Kelvin and Rossby waves propagation). McPhaden et al. [9] also showed that the Wyrtki jet is driven by westerly winds and illustrated the role of wind-forced equatorial waves in affecting the autumn Wyrtki jet. In fact, the co-occurrence of nIOD and La Niña in 2010 might have contributed to the current anomalies [35]. In August 2010, La Niña

was well established and developed quickly, resulting in massive westerly anomalies in the equatorial Indian Ocean. By September, nIOD occurred, strengthening the westerly anomalies [26]. The SMC in summer and Wyrtki jet in October, which flowed farther eastward, were probably caused by the westerly anomalies associated with La Niña and nIOD in 2010. In October 2010, strong westerly wind anomalies along the equator drove anomalous equatorial downwelling Kelvin waves propagating eastward, which reached the western coast of Sumatra and converted into coastal downwelling Kelvin waves [12]. These coastal downwelling Kelvin waves assisted in the development of the EICC, which was noted by Dandapat et al. [25]. The downwelling coastal Kelvin waves also reflected Rossby waves westward, which could be presented by salinity anomalies.

The co-occurrence of nIOD and La Niña in 2010 also played an important role in the variation in precipitation. The anomalous precipitation was above normal over western India but below normal over the northern BoB, which is likely associated with the nIOD in 2010 [15]. Additionally, the precipitation over most parts of India was above normal, which was probably due to the La Niña events [18]. Sankar et al. [46] also found that more moisture was concentrated over India in 2010. The moist static energy (MSE) anomalies at 850 hPa (Figure 12) confirmed that the atmospheric instability was beneficial to convection. The summer monsoon was depicted by MSE anomalies from June to August. However, from September onward, the unstable air mass was confined between 5°N and 25°N, suggesting more convection activity over India in addition to the region south of the Himalayas. In fact, the nIOD gave rise to westerly anomalies in the central Indian Ocean. Together with the strong summer monsoon associated with La Niña, this caused a convergence over India, which favored more freshwater transport to the east of Sri Lanka.

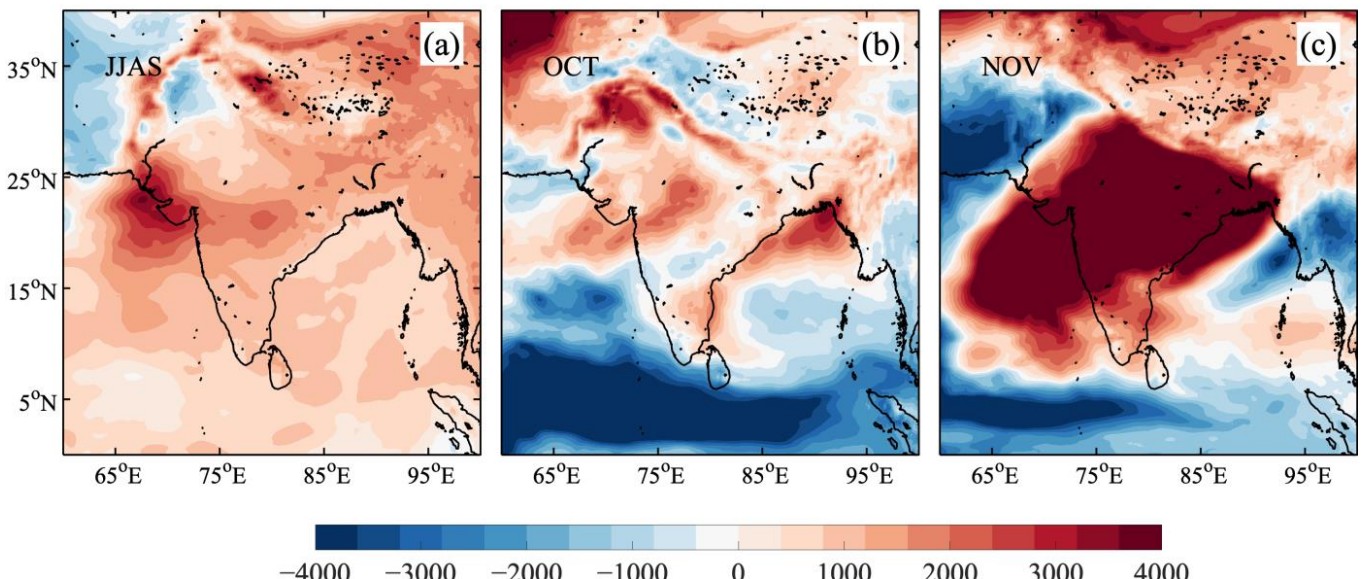

**Figure 12.** Moist static energy anomalies at 850 hPa (shaded; J/kg) (**a**) from June to September, (**b**) in October, (**c**) in November in 2010 based on ERA5 data.

The negative salinity tendency around Sri Lanka in autumn 2010 probably influenced the formation of the Arabian Sea Mini Warm Pool (ASMWP) in the spring of 2011, which was associated with Indian summer monsoon in 2011. From November 2010 to February 2011, the freshwater around Sri Lanka was transported to the southeastern AS (SEAS) by the west Indian coastal current (WICC) and mostly trapped in an anticyclonic eddy, strengthening the salinity stratification there (Figure 13c). Figure 13 presents the anomalies of salinity difference between the surface and 50 m and confirms the strong salinity stratification in the SEAS. The strong salinity stratification (Figure 13) was characterized by a shallow MLD. In addition, the westward downwelling Rossby waves triggered by coastal downwelling Kelvin waves gave rise to an anticyclonic eddy circulation known as

the "Lakshadweep high (LH)", which deepened the isothermal layer [47,48]. As a result, a thick barrier layer developed and reduced the heat exchange between the mixed layer and thermocline, which paved the way for the formation of the ASMWP in the following spring [49]. Furthermore, the thermal inversions embedded within the barrier layer also contributed to the formation of the ASMWP [50,51]. The ASMWP is located in the southeastern Arabian Sea (SEAS; 4–14°N, 68–78°E) and characterized by a sea surface temperature (SST) exceeding 30.25 °C [52]. The ASWMP is built up in March and dissipates in May [53]. With the light wind and clear sky in March, increased solar radiation influx is trapped in the thin mixed layer [50,54], resulting in rapid SST increases and the formation of the ASMWP [55].

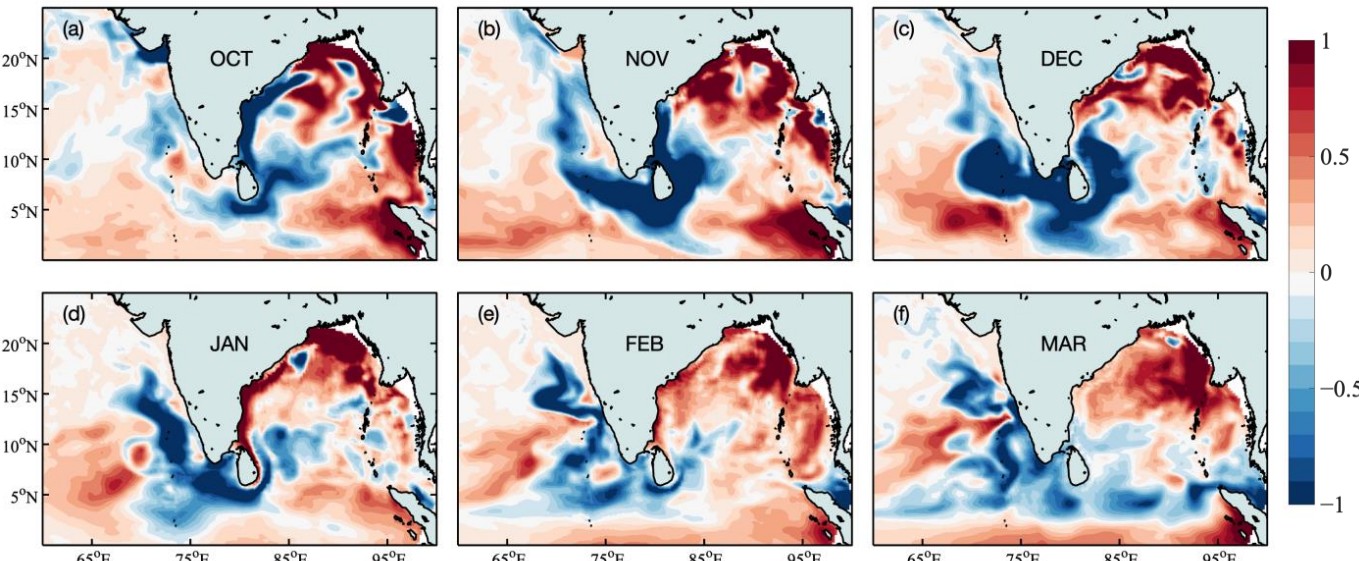

**Figure 13.** Monthly distribution of anomalies of salinity difference between surface and 50 m depth (shaded; psu) in (**a**) October 2010, (**b**) November 2010, (**c**) December 2010, (**d**) January 2011, (**e**) February 2011, and (**f**) March 2011 based on ORAS5 data.

Previous studies also pointed out that the ASMWP affects the Indian summer monsoon through its influence on the monsoon onset vortex (MOV), a low-pressure system over the SEAS [53,56]. An increase in moisture at mid-tropospheric levels resulting from the ASMWP leads to an increase in equivalent potential temperature and a positive low-level vorticity, which contributes to the surface convergence and divergence in upper levels. Furthermore, the weaker vertical shear of zonal winds favors the formation of the MOV [57]. The MOV often deepens into a cyclonic storm, followed by the strengthening of westerlies. The westerlies subsequently lead to sustained rainfall over Kerala, heralding the onset of the summer monsoon. Then, the MOV modulates the northward advancement of Indian summer monsoon [53]. A schematic summary of the processes associated with the salinity anomalies in 2010 is shown in Figure 14.

Based on these processes, with the co-occurrence of nIOD and La Niña, the massive low-salinity waters from Sri Lanka carried into the SEAS and the westward Rossby waves associated with the strength of LH in winter 2010 might have an effect on the formation and development of ASMWP, and thus influenced the onset and rainfall of the summer monsoon in 2011.

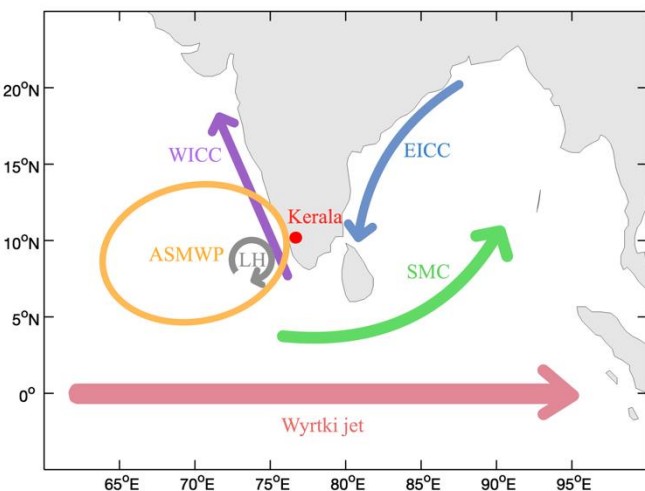

**Figure 14.** Schematic of the processes associated with the salinity anomalies in 2010. BoB is characterized by a low SSS while the Arabian Sea features a high SSS. The SMC (green arrow) in the summer monsoon season and the Wyrtki jet (pink arrow) in October flowed father eastward, causing more high-salinity water to move into the northern BoB in 2010. In addition, the EICC (blue arrow) was stronger in October 2010, resulting in extremely negative salinity anomalies around Sri Lanka and positive salinity anomalies in the northern BoB. Most of the freshwater was then transported to SEAS by WICC (purple arrow) and trapped by LH (gray arrow). Then, it could influence the formation of the barrier layer there in winter 2010 and the ASMWP (yellow circle) in spring 2011. Subsequently, the ASMWP affects the low-pressure system known as MOV over the SEAS. The MOV often deepens into a cyclonic storm followed by the strengthening of westerlies. The westerlies lead to sustained rainfall over Kerala (red dot), heralding the onset of the summer monsoon.

## 5. Conclusions

Positive salinity anomalies exceeding 2 psu in the northern BoB and negative salinity anomalies reaching -2 psu east of Sri Lanka in autumn 2010 were detected by the SMOS dataset and reanalysis products. Here, we analyzed the anomalous salinity budget and determined that anomalous horizontal advection played a dominant role in the salinity fluctuation within the mixed layer. In particular, the horizontal advection of climatological salinity by anomalous currents was confirmed to be critical to the salinity variations in both regions. The SMC in summer and Wyrtki jet in October, which flowed farther eastward, were probably caused by the westerly anomalies associated with La Niña and nIOD in 2010. In addition, there were anomalous northward currents in the eastern BoB. The strong SMC and Wyrtki jet combined with the anomalous basin-scale cyclonic circulation favored water transport to the northern coast of the BoB and led to the positive salinity tendency there. On the other hand, more runoff of the rivers (Mahanadi, Godavari and Krishna rivers) attributed to increased precipitation over eastern India in summer provided more freshwater for the EICC. The strengthened EICC related to the anomalous downwelling Kelvin waves carried local low-salinity waters southward and led to the negative salinity tendency on the eastern coast of Sri Lanka. Subsequently, the freshwater around Sri Lanka was carried farther into the southeastern Arabian Sea by the WICC in November, which was associated with local salinity stratification. The strong salinity stratification could influence the formation of the barrier layer in winter and the variation of the ASMWP in the following spring. The ASMWP could affect the ISM through its influence on the MOV over the SEAS. The probable causes of low-salinity water transported into SEAS and its influence on salinity stratification are preliminarily discussed in this study. Further research is needed to gain a clearer understanding of the association of the spatial pattern of salinity anomalies in the BoB with the air–sea interactions in the tropical Indian Ocean such as the formation of the ASMWP.

**Author Contributions:** Conceptualization, F.Q.; methodology, F.Q. and S.C.; validation, F.Q.; formal analysis, S.C. and F.Q.; data curation, F.Q. and S.C.; writing—original draft preparation, J.C. and S.C.; writing—review and editing, F.Q., J.C., C.J., Y.Q. and J.X.; visualization, S.C.; funding acquisition, F.Q. and Y.Q. All authors have read and agreed to the published version of the manuscript.

**Funding:** This work was supported by the Scientific Research Foundation of the Third Institute of Oceanography, MNR (Grant numbers: 2022021), the National Natural Science Foundation of China (42130406), the Global Change and Air–Sea Interaction II Program (GASI-04-WLHY-03, GASI-04-QYQH-01; GASI-01-SIND-STwin).

**Data Availability Statement:** The SMOS sea surface salinity is obtained at https://earth.esa.int/ (accessed on 1 November 2020). The ERA5 and ORAS5 datasets are obtained at https://cds.climate.copernicus.eu/ (accessed on 20 November 2020).

**Acknowledgments:** We would like to thank the reviewers for their time spent on reviewing our manuscript and their comments helping us improving the article.

**Conflicts of Interest:** The authors declare no conflict of interest.

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
