# Peer review of "Sea Surface Salinity Anomaly in the Bay of Bengal during the 2010 Extremely Negative IOD Event"

_remotesensing, doi:10.3390/rs14246242_

Round 1

Reviewer 1 Report (Previous Reviewer 2)

There are some previous concerns that the authors have not addressed. See attachments.

Author Response

Thank you for your decision and constructive comments on our manuscript!  We have carefully considered the suggestions and tried our best to improve the manuscript. Please see the response in the attachment. 

Reviewer 2 Report (New Reviewer)

The paper is generally well written (some minor grammatical changes are included in the pdf). The figures text is generally tiny (cannot be read at a zoom of 100%) and its size needs to be increased to a readable size.

I believe it is publishable after doing the small corrections suggested (see the attached pdf).

Author Response

Thank you for your decision and constructive comments on our manuscript!  We have carefully considered the suggestions and tried our best to improve the manuscript. Please see the response in the attachment. 

Reviewer 3 Report (New Reviewer)

This work discusses a model's portrayal of the extreme salinity anomalies in the North Indian Ocean in 2010. It may prove of interest to researchers working in that area, but to widen it's readership more effort is required in the Introduction. There is some interesting discussion on implications in the final paragraphs, but it would have helped if there had been an introductory figure showing the previous knowledge on WICC, EICC, Wyrtki Jet, SMC, LH, ASMWP, MOV and Kerala. This might be just one figure or a pair of panels showing "summer" and "autumn", ideally showing the changes expected with nIOD. A clearer pictorial representation of what is already known would help the general reader see the connections and help accentuate what is new in this work.

In most papers the meteorological seasons are 3 months long; this paper has a 4-month "summer" and 2-month "autumn". I know this is due to the strong monsoon forcing, but I think it would help general readers to clearly explain this choice.

The generally good agreement of SMOS and ORAS5 is nice to see, but I feel we need more detail on the statement that the model is perturbed by both observations and forcing fields (l. 123). If the model assimilates SMOS and other SSS observations then the similarity between model and satellites is not so remarkable or interesting. More detail is required on which salinity observations are used within the model.

I felt that Figures 1 & 3 should be larger (so easier to read) and could benefit from being made in colour. For other figures I felt the captions could be more informative, rather than requiring the reader to scan every label in a plot or the text a page or two earlier. For example:

Fig. 2 could add "a) June-September, b) October-November"

Fig. 4 plots anomalies relative to what time period. If these are the anomalies averaged over 2010-2019 they are showing a long-term trend relative to some other period, rather than a focus on 2010.

Fig. 5 shows plots for 2010 according to the text, but does not say so in the caption.

Fig. 7's caption refers to "summer" and "autumn", but one needs to look carefully at labels within the pictures to see left-hand is 3 months (not 4 months here?) of "summer" and then individual months of "autumn".

I was confused by the paper switching from freshwater anomalies to salinity anomalies -- surely one is the inverse of the other, so why switch arbitrarily between the two? Figure 9 was an extreme case showing the FW anomaliea across one section with the S anomalies across another, when one would like to compare the two.

In caption of Fig. 10 I think they mean "BoB" rather than "Indian Ocean".

l. 396-398 refer to all that FW around Sri Lanka being trapped in an eddy; surely the volume of water around Sri Lanka is much more than encompassed within one eddy?

Given that changes in E-P and River runoff were supposed to be small, shouldn't the salinity tendency almost equal the sum of the individual items of the freshwater budget? l.259-260 suggests that the anomalous salinity tendency is mainly attributable to advection, whereas PSPT in Fig. 6a seems to be ~20% of the sum of the advection terms, and for many months is of opposite sign. Thus I do not immediately have confidence in one term explaining the others. Your results may be correct, but I feel you need to more clearly argue the case in order to convince other readers.

Finally, I think a Hovmoller diagram of salinity anomalies along the west and east coasts of India, say at 50 km offshore, would help demonstrate the causal links.

Author Response

Thank you for your decision and constructive comments on our manuscript!  We have carefully considered the suggestions and tried our best to improve the manuscript. Please see the response in the attachment. 

Round 2

Reviewer 1 Report (Previous Reviewer 2)

The authors have addressed most of my major concerns, and the revised the manuscript deserve publication.

However, I encourage the authors to add the DOI for each reference in their reference list but no effort has been made in both rounds of revision. I think it is needed to be done according to format guide of the Remote Sensing. See instructions at https://www.mdpi.com/authors/references.

Author Response

Thank you for your decision and constructive comments on our manuscript!  We modified the format of the reference.

Reviewer 3 Report (New Reviewer)

Thank you for your response to my comments. I found your figure R1 very useful and was surprised you did not think it would help other readers not fully familiar with the locations of WJ, SMC, LH etc. However I would strongly implore you to add Figs S2 & S3 (maybe as one 2-part figure) to your final version. It is a lovely demonstration of how strong the signal remains going from A to B to the tip of Sri Lanka and then becomes somewhat weaker and less distinct (at the coast) from Kerala onwards.

The larger figures are generally much easier to read. Fig. 6 looks a lot better, much clearer and also better agreement of tendency and terms. However I still do not understand why the tendency looks like only 25% of the sum of the terms. Your response states that this is due to "processes such as vertical entrainment". Are you sure this is the case, or is it just a guess? I note in your response to Reviewer 1 you stated that you neglected the vertical advection and entrainment terms. I really feel you need to offer some explanation in the manuscript to enable other readers to understand.

I note your increased detail on the forcing and observations used in the model. It would be helpful if in the manuscript you could explicitly state "No observations from SMOS or other satellite measures of SSS are used in the model, so its output is independent from the SMOS observations." The clear demonstration of the independence of the model from SMOS would give much greater weight to the good agreement of SMOS and ORAS5 output.

A few minor typographical/grammatical errors:

Always use capital 'O' for 'Indian Ocean' (l.100, 109, 395, 484).

Insert a hyphen in "co-occurrence" (l.392, 405, 452).

The "where ..." clause after an equation should not normally be indented as it is part of the same paragraph (l.156, 184, 193).

l. 19 Delete "the" before "Sri Lanka".

l. 97 Delete "was"

l. 101 Delete first "2010" to read "distribution in summer and autumn 2010"

l. 133 Change "zonal (meridional)" to "zonal and meridional" and then remove last sentence of that paragraph.

Caption to Fig. 3 should read " ... in 2010 relative to the mean for the period 2010-2019."

l. 238 & 239 Change "lied" to "lay" and "sit" to "sat".

l. 267-271 The section "It should be noted ... v' dS'/dy))" is a repeat of what was in section 2.2.1 and so should be removed.

l. 282 Delete first "salinity" to read "advection of the climatological salinity"

l. 375 Change "explored" to "explore".

l. 390 Use a capital 'P' in "McPhaden".

l. 406 Change to "over western India"

l. 408 Change to "over most parts of India"

l. 413 Write "air mass" as two words.

l. 417 Insert "the" to read "to the east of"

l. 443 -4 Combine sentences to read "...monsoon onset vortex (MOV), a low pressure system over the SEAS [56,59]. An increase in ..."

Fig. 12 Change caption to read "... from October 2010 to March 2011 based on ..."

l. 471 Where you write "Indian Ocean" should this be "BoB", as it is not the case for AS?

l. 473 Delete "the" to read "over eastern India".

l. 474 Insert "the" to read "for the EICC.".

l. 476-480. The sentence from "Subsequently ..." to "... following spring" is too long, so break it into 2 or 3 sentences.

l. 481 Change "probably" to "probable"

Author Response

Thank you for your decision and constructive comments on our manuscript!  We have carefully considered the suggestions and tried our best to improve the manuscript. Please see the response in the attachment. 

This manuscript is a resubmission of an earlier submission. The following is a list of the peer review reports and author responses from that submission.

Round 1

Reviewer 1 Report

Review of “Sea Surface Salinity Anomaly in the Bay of Bengal During the 2010 Extremely Negative IOD Event”

In this study, the authors sought to investigate the variability of sea surface salinity (SSS) in the Bay of Bengal during the 2010 negative Indian Ocean Dipole (IOD) event. Their results suggest that the SSS variability was more influenced by the horizontal advective term. The impact of the SSS variability on barrier layer and Arabian Sea warm pool are also highlighted. There are some significant challenges with this manuscript which makes it unacceptable for publication in its current state. There is no clear sign of originality as many aspects of the results and analyses are already well known in the literature. There are a couple of unclarities and inconsistencies in the manuscript that need to be rectified. Some results and conclusions drawn are very speculative with no evidence provided to support them. I therefore suggest rejection of the manuscript.

  1. L23: “favored such transport”, not sure what this is referring to. Please clarify or modify the writeup.
  2. L32: spell out P and E.
  3. L45-46: please clarify, are you suggesting that equatorial Kelvin waves generate the Wyrtki Jets?
  4. L57-58: significant more than the IOD in the Indian Ocean?
  5. L71-76: need to elaborate more why the 2010 IOD is significant to study. It may help to include a plot of the dipole mode index
  6. L86: spell out SMAP
  7. L87-89: what variables did you obtain from ORAS5?
  8. L102: how did you obtain “mixed layer salinity” from SMOS?
  9. L103: what criterion was used to compute the mixed layer depth?
  10. L104-105: if both S and So are from ORAS5, then where was SMOS used in the budget estimation?
  11. L100: equation 1- what about the contribution of the subsurface, i.e., the interaction between the mixed layer and the layer below it?
  12. L117: why the need to calculate all these transports?
  13. L117-131: are these fluxes or transports? If transports in the mixed layer as stated, then they should be integrated over the mixed layer. Thus, a term (depth integral) is missing in the equations presented.
  14. L135-136: why not compute these as was done for the salt budget (i.e., equation 1)?
  15. L142: heat capacity of what?
  16. L142: temperature of what?
  17. L154: which high salinity water is being referred to here?
  18. L155-156: sentence here is confusing. Please revise
  19. L159-168: should this section not come before the L147-158?
  20. L160: you called them “region’ in L159, you should stick to that and not “sector”
  21. L165: how does surface currents make salinity lower in BoB?
  22. L162-168: the write up here is very confusing. Are you comparing SSS from SMOS and ORAS5 or SSS at the two locations?
  23. L166-168: the basis for the conclusion is quite weak. This needs to be revised with more rigorous basis to draw such conclusion
  24. L181-183: what is the basis for this conclusion? No currents are overlaid to support this statement.
  25. L185: what do you mean by less significant?
  26. L187-189: this conclusion is speculative. No evidence has been provided to support this
  27. L191: monthly averaged? These appear to be anomalies (i.e., based on the color bar). If so, are they seasonal or interannual anomalies?
  28. Please see above
  29. L196-204: This section needs to be further developed. It may help to elaborate on the changes to the terms as the seasons progress. For example, 5a shows low SSS in NE India, which then advects down along the coastal rim to southern Sri Lanka and then into the AS.  There are also a couple of interesting features in Fig 5m-r (advective term) that needs to be developed/discussed.
  30. Figure 7: it is hard to see the vectors. It may help to plot fewer vectors that are larger.
  31. L246-261 and Figure 8: this is confusing. Not sure one you are investigating freshwater transport in one section and saltwater transport in another. Freshwater is just 1-saltwater that is scaled by a factor/variable.
  32. L270-271: climatological winds are not presented to aid this comparison
  33. L267: section needs to be better presented to enhance comprehension of role in SSS variability during the 2010 IOD event

Reviewer 2 Report

The paper “Sea Surface Salinity Anomaly in the Bay of Bengal During the 2010 Extremely Negative IOD Event” investigated the causes of salinity anomalies in autumn 2010 over the northern Bay of Bengal and around Sri Lanka. The authors attribute the salinity anomalies to advection, which is associated with the summer monsoon current, Wyrtki jet and east Indian coast current. While the topic is of interest to the ocean community, I believe it requires substantial improvements before being published.

In particular, the methods section needs improvement, and the definition of the critical variables (how anomalies are derived) or properties (how summer is defined in this manuscript) needs to be clear. A comparison with previous related studies should be presented in the discussion. The novelty of this analysis should be highlighted. Evidence is lacking to support some of the conclusions.

I find major revisions are needed. The authors should address the following concerns to be deserved for publication.

Details can be found in the attachment.
